# A Cross-Sectional Study of the Associations between Chronotype, Social Jetlag and Subjective Sleep Quality in Healthy Adults

**Sudha Raman and Andrew N. Coogan *** 

Department of Psychology, Maynooth University, National University of Ireland, County Kildare, Ireland
* Correspondence: andrew.coogan@mu.ie; Tel.: +353-708-6624

**Abstract:** Social jetlag, a mismatch between internal biological time and social schedules, and a later timing of the midpoint of sleep on work-free days as an indicator of the circadian phase of entrainment (late chronotype), may be factors associated with poor quality sleep. This study examined the association of social jetlag and chronotype with subjective sleep quality ratings in a healthy young adult cohort and interrogated the moderating effects of sex and age on these associations. A total of 1322 participants aged 18 to 40 completed the Pittsburg Sleep Quality Index (PSQI) and the Munich Chronotype Questionnaire. Later timing of midsleep on "free" days (an indicator of chronotype) had a small-to-medium association with poorer subjective sleep quality, independently of sex and age (rho = 0.212, $P < 0.001$). Greater social jetlag had a small association with poorer subjective sleep quality ratings (rho = 0.077), and this effect was moderated by sex with there being a relationship between social jetlag and sleep quality only in males. Social jetlag did not mediate the relationship between chronotype and sleep quality. These results indicate differential relationships of the chronotype and social jetlag with subjective sleep quality and indicate that sex is a moderating factor for sleep quality's relationship with social jetlag, but not for the association between sleep quality and chronotype.

**Keywords:** social jetlag; chronotype; sleep quality; circadian

## 1. Introduction

In sleep research, much attention is paid to the constructs of sleep duration and sleep quality. Good quality sleep is indicated by factors such as short sleep latency, greater sleep efficiency and fewer awakenings after sleep onset [1]. Sleep is fundamentally a biological process driven by homeostatic and circadian processes [2]; these biological drivers of sleep are, however, subject to extrinsic societal forces, and as such, understanding how homeostatic and circadian processes influence sleep quality directly and interact with social factors will increase our understanding of the determinants of sleep quality [3].

Chronotype may refer to actual and/or preferred timing of sleep/wake behaviours [4], and individuals with evening preference/later chronotype may experience impaired sleep quality [5,6]. Similarly, social jetlag (SJL), the manifestation of differences in sleep timings on "free" and "work" days arising as a conflict between social schedules and the internal circadian clock [7], may also be associated with impaired sleep quality [8]. Furthermore, chronotype is influenced by sex and changes through the lifespan (being male, and an adolescent or young adult, are associated with later chronotype) [9]. Greater SJL is most common with later chronotype [7], and therefore, by extension, may be influenced by age and sex as being males and in the age range of the late teens to early twenties is associated with both greatest SJL and later, chronotype [7]. Subjective sleep quality also decreases with increasing age [10]. In this study, we examined the associations between chronotype and SJL with subjective sleep

quality (assessed by the Pittsburg Sleep Quality Index (PSQI)) in a healthy young adult population and examined whether these associations were moderated by sex and/or age. We hypothesized that both timing of midsleep on free days (MSFsc; a key measure of chronotype) and SJL would be associated with PSQI score, and these relationships would be influenced by age and sex.

## 2. Results

The distribution of MSFsc (mid-sleep on free day), SJL and PSQI scores, by age group and sex, is shown in Figure 1A,B, and descriptive statistics are presented in Table 1. The age dependence of MSFsc, SJL and PSQI scores are shown in Supplementary Figure S1.

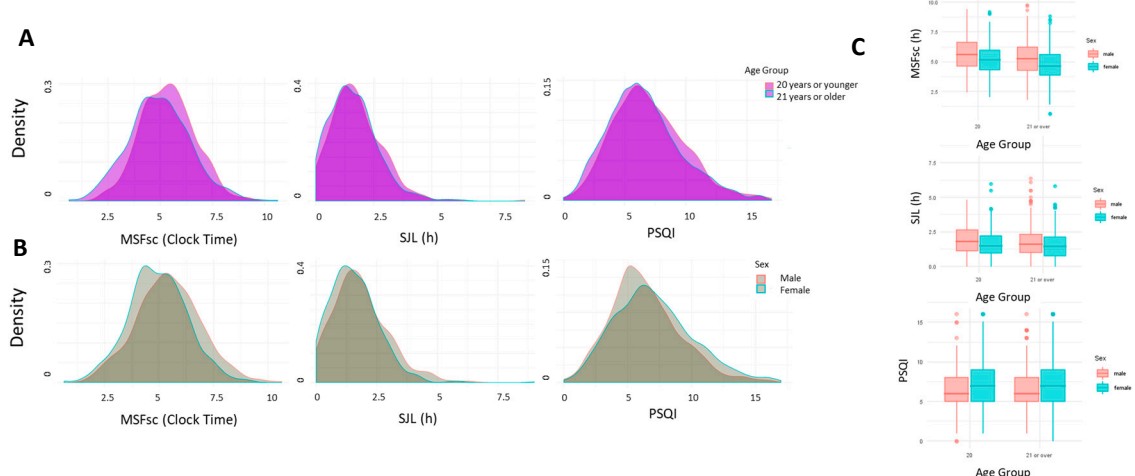

**Figure 1.** Density plots showing the distribution of MSFsc, SJL and PSQI scores in (**A**) participants 20 years old or younger, and those of 21 years or older and (**B**) males and females. (**C**) Box plots showing the main effects of sex and age on MSFsc and SJL, and sex on PSQI, but no age group and sex interactions effects on MSFsc, SJL or PSQI.

**Table 1.** Descriptive statistics of the study cohort.

| **Male** *N* = 522, 39.5% **Mean Male Age** = 22.9, SD = 4.8 | | **Female** *N* = 799, 60.5% **Mean Female Age** = 22.2, SD = 4.6 | |
|---|---|---|---|
| **PSQI Good Sleepers** *N* = 305, 23.1% | | **PSQI Bad Sleeper** *N* = 1017, 76.9% | |
| | **Mean** | **Median** | **SD** |
| **Age (yrs)** | 22.66 | 21 | 5.26 |
| **MSFsc (h)** | 5.13 | 5.11 | 1.41 |
| **SJL (h)** | 1.68 | 1.51 | 1.05 |
| **PSQI** | 6.98 | 6 | 2.95 |

Factorial ANOVA using sex and age (dichotomized as 20 years or younger and 21 years or older, operationalized as such as there is an inflection point for late chronotype in the early 20s [11,12], (males 36% in the younger group, 42% of the older group), revealed a number of effects (Figure 1C). There were significant main effects of age group ($P < 0.001$, partial ETA squared = 0.021) and sex ($P < 0.001$, partial ETA squared = 0.032) on mean MSFsc, but no interaction between sex and age group ($P = 0.92$). Likewise, there were effects of age group ($P < 0.001$, partial ETA squared = 0.005) and sex ($P < 0.001$, partial ETA squared = 0.014) on mean SJL, but no sex and age group interaction ($P = 0.76$). Finally, there was an effect of sex on mean PSQI scores ($P < 0.001$, partial ETA squared = 0.011), and a marginal effect of age group ($P = 0.042$) and no age group and sex interaction ($P = 0.863$).

The correlation analysis shows a moderate positive relationship between MSFsc and PSQI scores (later MSFsc associating with poorer quality sleep; rho = 0.212, Bootstrap 95% CI lower bound = 0.157,

upper bound = 0.263, *P* < 0.001; Figure 2A,B). There was a statistically significant, but weak association between SJL and PSQI (rho = 0.077, Bootstrap 95% CI lower bound = 0.02, upper bound = 0.136, *P* = 0.005; Figure 2A,B). Neither SJL or MSFsc are correlated with average weekly sleep duration (rho = −0.074, *P* = 0.052 and rho = −0.034, *P* = 0.32 repetitively), and inclusion of average sleep duration as a covariate did not alter the relationships between SJL/MSFsc and PSQI. There was an expected positive strong correlation between SJL and MSFsc (rho = 0.468, Bootstrap 95% CI lower bound = 0.42, upper bound = 0.514, *P* < 0.001). The potential confounding effect of age on the relationships between MSFsc/SJL and PSQI were examined with partial correlations: the relationship between MSFsc and PSQI persisted when controlling for age (*r* = 0.202, *P* < 0.001), whilst the relationship between SJL and PSQI also persisted when controlling for age (*r* = 0.060, *P* = 0.03). When moderation analyses were carried out to include sex as a potential moderator between MSFsc or SJL and PSQI scores, it was found that the relationship between MSFsc and PSQI was not moderated by sex (*P* = 0.52 $R^2$ = 0.069 for the model; Figure 2). When the moderating effect of sex on the relationship between SJL and PSQI score was examined ($R^2$ of the model = 0.0228), there was a significant moderating effect of sex (P = 0.032). Upon further examination, it was revealed that the relationship between SJL and PSQI was only statistically significant in males (*r* = 0.184, *P* < 0.001) and not in females (*r* = 0.029, *P* = 0.42). A mediation analysis revealed SJL did not mediate the relationship between MSFsc and PSQI scores, indicating that SJL did not account for the association between MSFsc and PSQI score (Figure 2C).

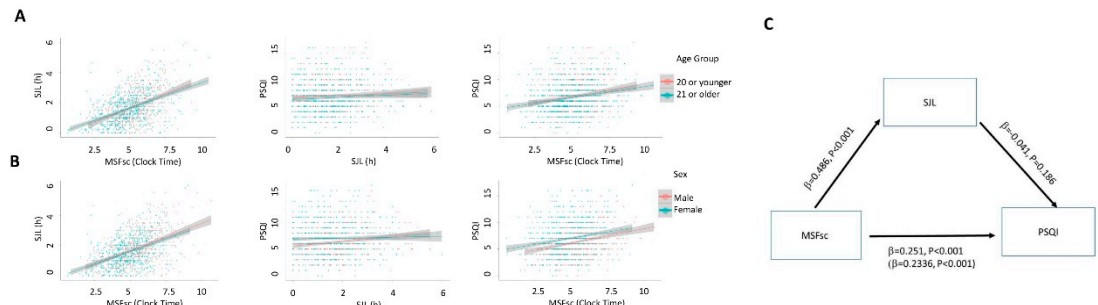

**Figure 2.** Scatterplots showing the associations between MSFsc, SJL and PSQI (**A**) in younger and older participants and (**B**) in males and females. (**C**) Mediation analysis of the relationship between MSFsc and PSQI by SJL.

## 3. Discussion

Our results indicate differential relationships between subjective sleep quality and chronotype or SJL. Later chronotype is associated with poorer sleep quality, and this association is independent of sex and age. Greater SJL is also associated with poorer sleep quality, but this association is weaker than that with chronotype and on sub-group analysis, is observed only in younger males. There are suggestions in the literature that sex may moderate the effects of SJL on various health parameters; for example, Cespedes Feliciano et al. [13] recently demonstrated that adiposity was associated with social jetlag in adolescent girls but not boys. It is not currently clear what mechanisms might underpin the differential sex effects of SJL on various psychological and physiological outcomes, but hormonal differences and gendered differences in affective and other psychosocial parameters are plausible factors of interest. The overall relationship between SJL and PSQI score in our study is small, but this may be accounted for by the lack of association in females. Furthermore, it has been previously noted that shorter sleep shows only a minor association with PSQI scores [14]; in the current dataset, average sleep duration measured via the MCTQ did not account for the relationship between SJL and PSQI scores.

Another important consideration for the interpretation of the current findings is the report of Pilz and colleagues (2018) [3] demonstrating that PSQI scores reflect sleep quality on workdays and not on free days during which sleep quality was improved. Furthermore, the difference between workday and freeday PSQI scores were related to chronotype (evening types having greater workday/free day discrepancies), and this effect was mediated through SJL. The current study did not differentiate

between PSQI scores on work and free days, and it will be of clear interest for future studies to do so. As SJL does not mediate the association of later chronotype with poorer quality sleep in our sample, it may be that other chronotype-associated traits and states mediate this relationship; for example, late chronotype is associated with more depression and anxiety symptoms, which, in turn, are associated with poorer quality sleep [15,16].

*Strengths and Limitations*

There are a number of important caveats that frame the interpretation of the current results. The first is that the sample was a "convenient" one, and as such, there is risk of sampling bias and consequent potential limitation of generalizability. Furthermore, the gender composition of the sample is biased towards females, and as such, the interpretation of the results should be nuanced in this context. One important feature of the current study is that the data were collected in the same season (late autumn–winter), and the potential confound of seasonal changes in sleep timing and quality should not be in play. However, the period of data collection also included the seasonal clock change at the end of October, a factor which may have impacted, in particular, ratings of SJL. An important caveat to the interpretation of the study is that the PSQI provides an estimate of sleep over a course of a month as perceived by the participants, and some studies have shown that subjective PSQI scores do not correlate with objective actimetric measures of sleep [17]. Furthermore, future work exploring the associations between chronotype, social jetlag and sleep quality in other age groups (e.g., children or older adults) would shed important light on developmental trajectories of such associations.

## 4. Materials and Methods

Participants aged between 18 and 75 and residing in Ireland completed the questionnaires ($N = 1466$ with complete responses) as part of a number of final year undergraduate research projects in psychology investigating chronobiological influences on a number of psychological processes. Participants were either students of Maynooth University (53.3813° N, 6.5918° W) or were acquaintances of students of Maynooth University. Data were collected between September and December 2012–2017, and pencil and paper versions of the questionnaires were used. Ethics approval was from the Maynooth University Research Ethics Committee intra alia the Department of Psychology's Research Committee. Recruitment was by convenience sampling via flyers, emails or personal contacts. Respondents were non-shift workers and did not report any significant health issues. To limit the scope of the analysis to younger adults, only respondents 18–40 years of age were included in the final analysis. This yielded a sample size of 1322 who had complete responses (male = 39.5%; 18–25 years old = 84%; mean age = 22.66). Subjective sleep quality was assessed using the Pittsburg Sleep Quality Index (PSQI) [18], a 19-item self-report tool. The total score was used as a continuous measure and a cutoff score of more than five indicates poor sleep quality and this was used to form good and poor sleep groups. The Munich Chronotype Questionnaire (MCTQ) [19] was used to assess the timing of actual sleep-wake behaviour and the circadian phase of entrainment. The key output is the time of mid-sleep (halfway between sleep onset and offset) on work (MSW) and work-free days (MSF). Sleep-corrected MSF (MSFsc) was used to estimate chronotype, to remove compensatory sleep due to accumulated sleep debt [20]. Social jetlag was calculated as the absolute difference between MSW and MSF [20]. All participants completed the questionnaires in pen and paper form.

Statistical analyses were conducted using SPSS. The SPSS PROCESS macro (version 3.3 [21]) was used to perform simple moderation analyses (model 1) with chronotype and SJL (continuous measures) as independent variables, and for mediation analysis (with SJL as the mediator between MSFsc and PSQI, PROCESS model 4). Age was treated as a continuous variables and also dichotomoised to produce a categorical variable for ANOVAs. Correlations (Spearman Rho), partial correlations and ANOVAs were run using Bootstrapping operations with 1000 iterations. Collinearity between MSFsc, SJL and PSQI was inspected using VIF values which revealed no significant multi-collinearity (VIF values of 1.4, 1.3 and 1.1 respectively). $P < 0.05$ was taken as indicating a statistically significant effect

for non-parametric correlations, and $P < 0.01$ for ANOVAs to account for the non-normal distributions of SJL and PSQI. Effect sizes were calculated as partial ETA squared values and interpreted according to Cohen (1988) [22].

**Supplementary Materials:** The following are available online at http://www.mdpi.com/2624-5175/2/1/1/s1, Figure S1: Distribution of MSFsc, SJL and PSQI by age in the sample.

**Author Contributions:** Conceptualization, formal analysis, data curation, A.N.C.; writing—original draft preparation, S.R.; writing—review and editing, A.N.C. and S.R. All authors have read and agree to the published version of the manuscript.

**Funding:** S.R. is funded by the John and Pat Hume Doctoral Scholarship Fund from NUI Maynooth University. This research received no other external funding.

**Acknowledgments:** Final year undergraduate students of the Department of Psychology helped with the data collection as part of their final year projects.

**Conflicts of Interest:** The authors declare no conflict of interest. The doctoral scholarship funding had no role in the design of the study; in the collection, analyses, or interpretation of data; in the writing of the manuscript, or in the decision to publish the results.

## Abbreviations

| | |
|---|---|
| SJL | Social Jetlag |
| PSQI | Pittsburg Sleep Quality Index |
| MCTQ | Munich ChronoType Questionnaire |
| MSW | Mid-Sleep Workday |
| MSFsc | Mid-Sleep Free day, sleep corrected |
| ETA | SPSS statistics for effect size |

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
