# Peer review of "A Cross-Sectional Study of the Associations between Chronotype, Social Jetlag and Subjective Sleep Quality in Healthy Adults"

_2624-5175, doi:10.3390/clockssleep2010001_

Round 1
Reviewer 1 Report
The submission is a Brief Report, and it does cover an important and not well understood topic; but I fear that the submitted manuscript is too brief in writing, leaving room for misunderstandings with also some statistical shortcomings that should be addressed. My comments are listed below.
1) Introduction: The rationale of including sex in the statistical model is unclear in the Introduction, the Introduction should introduce the reader to the content to understand the steps taken in the analysis and discussion. The authors write (line 42) "and therefore by extension may be influenced by age and sex.", but leave out to justify this decision.
2) Social jetlag appears (figure 1) better represented by the median instead of the mean, because of the right scewed distribution. The mean in this case is rather imprecise, compared to the median.
3) Social jetlag is not high in this population, as can be seen from figure 1, the density curve is wider for the younger participants compared to the older participants. This wider distribution in the younger individuals leads to a wider variance which allows for a higer probabilty of finding significant associations in the statistical models. Because overall social jetlag is rather low in this population, and the median somewhere close to 1 hour, I am not surprised to see the results as presented by the authors. I suggest the authors in their discussion to comment on these density differences between the age groups. Also see point 4.
4) Page 3, line 57: why has age 25 been used as a cut-off, when citing a paper that showed an earlier inflection point around the ages of 21?
5) Page 3, line 58: I am not convinced that the authors can say that there were "significant main effects of age group" on chronotype, since the ANOVA tests for group mean differences, giving a result that rather shows that the mean chronotype was different between the two age groups. If age effects on chronotype cannot be answered from this analyses. I suggest to rephrase. Same in the following sentences on social jetlag and PSQI.
6) Page 3, line 68: The authors entered age as a moderator to the correlation analysis between chronotype, social jetlag and PSQI. Age here cannot be treated as a moderator, because age both effects chronotype (hence also social jetlag) and sleep quality. Age is a confounder, not a moderator. Entering age here to the model, thereby, eliminates the effect, since age is connected to both outcome and exposure variable. This explains the null finding.
7) Discussion: The authors find an effect of sex on the outcome variables, but miss to discuss the reason how sex might play a role in this interplay. The authors cite a study on the association between sex, social jetlag and adiposity, but also here do not comment on explanatory mechanisms. How come that sex differentiates the findings here?
8) Methods: The fact that this study population was a "convenient sample" (and not a random sample) should be listed as a weakness in the discussion.
9) Methods: The authors write "To limit the scope of the analysis to young adults, only respondents 18-40 years of age were included in the final analysis." but do not give a reason for that step. The cut-off of 40 years appears arbitrary, especially since this is a convenient sample. Common statistical approaches in such case would be to draw a random sample from the total population, i.e. by means of bootstrap. In that line, how do results change when running the analysis on the full data set?
10) Methods: Please add how data was collected, i.e. online or paper pencil questionnaires?
11) Methods: Please provide more details on the statistical analyses, especially what models have been used, ie. linear regression models?
Author Response
Introduction: The rationale of including sex in the statistical model is unclear in the Introduction, the Introduction should introduce the reader to the content to understand the steps taken in the analysis and discussion. The authors write (line 42) "and therefore by extension may be influenced by age and sex.", but leave out to justify this decision.
Author’s Reply: The influence of sex and age on circadian and sleep factors hopefully has been made more clear now (lines 42-45: Greater SJL is most common with later chronotype [7], and therefore by extension may be influenced by age and sex as being males and in the age-ranges of the late teens to early twenties is associated with both greatest SJL and later chronotype. Subjective sleep quality also decreases with increasing age [11]. Clearly, a more in-depth discussion of the influences of age and sex on chronotype and sleep quality is beyond the scope of an introduction to a short communication.
2) Social jetlag appears (figure 1) better represented by the median instead of the mean, because of the right scewed distribution. The mean in this case is rather imprecise, compared to the median.
Authors’ Reply: In table one we have now given the median and mean scores for SJL, MCTQ and PSQI. Whilst SJL is skewed (as a function of its calculation as an absolute value), it is interesting to note that the median does not differ greatly from the mean in our sample (median of 1.51h versus a mean of 1.68 hours)
3) Social jetlag is not high in this population, as can be seen from figure 1, the density curve is wider for the younger participants compared to the older participants. This wider distribution in the younger individuals leads to a wider variance which allows for a higer probabilty of finding significant associations in the statistical models. Because overall social jetlag is rather low in this population, and the median somewhere close to 1 hour, I am not surprised to see the results as presented by the authors. I suggest the authors in their discussion to comment on these density differences between the age groups. Also see point 4.
Authors’ Reply: The SJL characteristics of the sample are very similar to what has been reported by us and others for this type of sample (median ~1.5h). With regards to the point about the width of the distributions, this is a valid point; however, following this reviewer’s and the second reviewer’s point about the age split, we reanalyzed the data with a dichotomization at 21 years of age. When the distributions are examined using this split, there is not a notable difference in the width of the density plots for SJL across the two age groups (Figure 1).
4) Page 3, line 57: why has age 25 been used as a cut-off, when citing a paper that showed an earlier inflection point around the ages of 21?
Author’s Reply: We have considered this point carefully, and have no modified our analysis to split the age groups at 21 years of age. Changing the age split does not markedly affect the results of the study (Figures 1 and 2). We also now include in Supplementary Figure one the age profile of MSFsc, SJL and PSQI in our sample.
5) Page 3, line 58: I am not convinced that the authors can say that there were "significant main effects of age group" on chronotype, since the ANOVA tests for group mean differences, giving a result that rather shows that the mean chronotype was different between the two age groups. If age effects on chronotype cannot be answered from this analyses. I suggest to rephrase. Same in the following sentences on social jetlag and PSQI.
Authors’ Reply: we have changed the phrasing to note main effects of group membership on mean values (lines 64, 66, 67)
6) Page 3, line 68: The authors entered age as a moderator to the correlation analysis between chronotype, social jetlag and PSQI. Age here cannot be treated as a moderator, because age both effects chronotype (hence also social jetlag) and sleep quality. Age is a confounder, not a moderator. Entering age here to the model, thereby, eliminates the effect, since age is connected to both outcome and exposure variable. This explains the null finding.
Authors’ Reply: We have removed age as a moderator in this analysis, and rather have used it a co-variate in partial correlations for the relationships between MSFsc/SJL and PSQI. Insertion of the age as a covariate in these analyses did not eliminate or substantially alter the previously described relationships (lines 77-80).
7) Discussion: The authors find an effect of sex on the outcome variables, but miss to discuss the reason how sex might play a role in this interplay. The authors cite a study on the association between sex, social jetlag and adiposity, but also here do not comment on explanatory mechanisms. How come that sex differentiates the findings here?
Authors’ Reply: We have now inserted this sentence to address this comment (lines 99-102): “It is not currently clear what mechanisms might underpin the differential effects of SJL on various psychological and physiological outcomes, but hormonal differences and gendered-differences in affective and other psychosocial parameters are plausible factors of interest.”
8) Methods: The fact that this study population was a "convenient sample" (and not a random sample) should be listed as a weakness in the discussion.
Author’s Reply: We have formulated a “Strengths and Weaknesses” section of the discussion , and address the “convenient” nature of the sample as a weakness (lines119-122): “There are a number of important caveats that frame the interpretation of the current results. First is that the sample was a “convenient” one, and as such there is the risk of sampling bias and consequent potential limitation of generalizability”
9) Methods: The authors write "To limit the scope of the analysis to young adults, only respondents 18-40 years of age were included in the final analysis." but do not give a reason for that step. The cut-off of 40 years appears arbitrary, especially since this is a convenient sample. Common statistical approaches in such case would be to draw a random sample from the total population, i.e. by means of bootstrap. In that line, how do results change when running the analysis on the full data set?
Authors’ Reply: We did indeed run the analysis on the full data set that had complete responses, and there were no substantive differences in the results. We limited the age range at 40 as the coverage of ages greater than 40 becomes very limited in our sample. We have also re-analysed our data using Bootstrapping, and this now allows us to report the upper and lower 95% CI bounds of the correlation analysis in our results
10) Methods: Please add how data was collected, i.e. online or paper pencil questionnaires?
Authors’ Reply: We have now clarified that data was collected via pencil and paper (line 139).
11) Methods: Please provide more details on the statistical analyses, especially what models have been used, ie. linear regression models?
Authors’ Reply: We do realise that our original description of the statistical procedures was unhelpfully sparse. We have now expanded the section to include sufficient details of the procedures applied (lines 159-165).
Reviewer 2 Report
Journal: Clock & Sleep
Type of report: Short Communication
Title: A Cross Sectional Study of the Associations between Chronotype, Social Jetlag and Subjective Sleep Quality in Healthy Adults
Authors: Sudha Raman and Andrew N. Coogan
The authors presented extensive data on the sleep-wake rhythm and sleep of young people in Ireland. They analyzed the relationship between chronotype, social jetlag, and sleep quality assessed with MCTQ and PSQI. To assess the relationship between the studied indicators, modern methods of statistical data analyses were used, which allowed to obtain some new data of interest to specialists in this field. In connection with the great importance of the topic raised by the authors in the manuscript, I suggest the authors to submit a detailed description of the collected material in the form of a full-size article.
I have some comments, mainly concerning the methodological part and the results that the authors need to take into account when submitting a revised manuscript.
At the end of the introduction section, hypotheses should be formulated that the authors would like to test in their study. Judging by the graph in figure 1, SJL has an asymmetric distribution. Was this fact taken into account when using this indicator in statistical analysis? Have the variables been analyzed for multicollinearity before being included in the model? It is necessary to use the abbreviations adopted by the authors throughout the text. Line 89-90. The authors point to a possible link between SJL, duration and quality of sleep, but did not provide their own data on sleep duration, although MCTQ and PSQI provide an opportunity to measure this indicator. Line 91-100. The authors discuss the possible impact of day of the week on sleep quality. They could conduct their own analysis of the association of weekend and workday sleep characteristics assessed by MCTQ with sleep quality and its specific characteristics (sleep duration, start and end of sleep, sleep latency) assessed by PSQI. There is no limitation section in the manuscript, although, as will be shown later, there are a number of shortcomings in the work. 84% of the examined persons were included in the age group of 18-25 years. It is not clear from the manuscript why exactly the age of 25 years was used by the authors as a point of division into 2 age groups. It is necessary to show on the graph the dependence of the chronotype, SJL and sleep quality on age in order to justify such a division. In addition, these data are of independent importance for comparing regional differences in age-related changes in these indicators; in recent years, such data are widely represented. There is too much bias in the gender composition of the examined persons towards the female sex. The authors need to indicate this in the limitations section. It is necessary to indicate how men are distributed by age groups. The authors give a very meager characterization of the examined individuals, which significantly impairs the quality of the collected material. What social groups were examined: students, workers, employees? At what time they begin work / study (the value of SJL very much depends on this indicator)? City / village, population, latitude. The authors, as an advantage of this study, indicated that the study was conducted in September-December, which, according to the authors, excludes the influence of the season of the year on the studied characteristics, but did not take into account that in October, the time is shifting from summer to winter time. This fact has a significant impact on the chronotype and SJL. It is necessary to take this factor into account in the analysis. It is necessary to describe the statistical processing section in more detail. It is necessary, in particular, to indicate whether age was used as a continuous or categorical value?
Author Response
At the end of the introduction section, hypotheses should be formulated that the authors would like to test in their study. Judging by the graph in figure 1, SJL has an asymmetric distribution. Was this fact taken into account when using this indicator in statistical analysis?
Authors’ Reply: We have now introduced a section setting out the hypotheses to be tested (lines 48-50): “ We hypothesized that both MSFsc and SJL was be associated with PSQI score, and these relationships would be influenced by age and sex.” SJL is indeed asymmetrically distributed (due in part to it being calculated as an absolute variable). We account for this non-normallity in the thresholds for significance for factorial ANOVAs and in the use of non-parametric correlation proceudres (outlined now in the methods section, lines 150-155.
Have the variables been analyzed for multicollinearity before being included in the model?
Authors’ Reply: Yes. VIF values were inspected for detection of mulicollinearity, and these values are reported now on line 161 (all VIF values were <1.5, indicating that multicollinearity was not a significant issue).
It is necessary to use the abbreviations adopted by the authors throughout the text.
Authors’ reply: This has been corrected.
Line 89-90. The authors point to a possible link between SJL, duration and quality of sleep, but did not provide their own data on sleep duration, although MCTQ and PSQI provide an opportunity to measure this indicator.
Authors’ Reply: We examined the relationship between SJL and average weekly sleep duration from the MCTQ, and find only very small relationships. Further, when we insert average weekly sleep duration as a covariate, relationships between SJL/MSFsc and PSQI are not altered (line 73-75). Therefore, we have altered the sentence in the discussion to reflect this finding: “Further, it has been previously noted that shorter sleep shows only a minor association with PSQI scores [12]; in the current data set, average sleep duration measured via the MCTQ did not account for the relationship between SJL and PSQI scores.” (lines 103-105).
Line 91-100. The authors discuss the possible impact of day of the week on sleep quality. They could conduct their own analysis of the association of weekend and workday sleep characteristics assessed by MCTQ with sleep quality and its specific characteristics (sleep duration, start and end of sleep, sleep latency) assessed by PSQI.
Authors’ Reply: Our point here is that the study of Pilz et al (2018) indicate that PSQI scores reflect workday sleep quality predominantly when the PSQI is modified to address work days separate to free days. That approach was not deployed in the current study, but clearly is of interest for future studies. We have examined MSF and MSW and their relationships to PSQI and do not find anything markedly different to the relationships for MSFsc (and so have not included the data).
There is no limitation section in the manuscript, although, as will be shown later, there are a number of shortcomings in the work. 84% of the examined persons were included in the age group of 18-25 years. It is not clear from the manuscript why exactly the age of 25 years was used by the authors as a point of division into 2 age groups. It is necessary to show on the graph the dependence of the chronotype, SJL and sleep quality on age in order to justify such a division. In addition, these data are of independent importance for comparing regional differences in age-related changes in these indicators; in recent years, such data are widely represented.
Authors’ response: We have now inserted an explicit “Strengths and Weaknesses” section of the discussion. As noted in our response to reviewer one, we have re-examined the data using an age-group split at 21, in accordance with the age of the inflection of chronotype described to date in various studies. As such, we have re-run our analysis and updated figures 1 and 2 accordingly. We have also included graphs of the age-dependence of MSFsc, SJL and PSQI in Supplementary Figure 1.
There is too much bias in the gender composition of the examined persons towards the female sex. The authors need to indicate this in the limitations section.
Author’s Response: This is now noted as a limitation (lines121-123: “Further, the gender composition of the sample is biased towards females, and as such the interpretation of the results should be nuanced in this context.”).
It is necessary to indicate how men are distributed by age groups.
Authors’ Response: We now give the mean and SD of age by sex in Table 1 and in line 62 give the percentage of males in each of the age groups.
The authors give a very meager characterization of the examined individuals, which significantly impairs the quality of the collected material. What social groups were examined: students, workers, employees? At what time they begin work / study (the value of SJL very much depends on this indicator)? City / village, population, latitude. The authors, as an advantage of this study, indicated that the study was conducted in September-December, which, according to the authors, excludes the influence of the season of the year on the studied characteristics, but did not take into account that in October, the time is shifting from summer to winter time. This fact has a significant impact on the chronotype and SJL.
It is necessary to take this factor into account in the analysis. It is necessary to describe the statistical processing section in more detail. It is necessary, in particular, to indicate whether age was used as a continuous or categorical value?
Authors’ Response: We give more details on the characteristics of the participants; however we did not collect addresses or work details beyond those collected via the MCTQ. We now note that the period of data collection would include the clock change to winter time in October, and the potential effects this may have (line 125). We hope it is clear now where age was treated as a continuous variable, and where it was treated a categorical grouping variable.
Round 2
Reviewer 1 Report
All points sufficiently addressed.
Reviewer 2 Report
No comments.